# [Cu(NN_1_)_2_]ClO_4_, a Copper (I) Complex as an Antimicrobial Agent for the Treatment of Piscirickettsiosis in Atlantic Salmon

**DOI:** 10.3390/ijms25073700

**Published:** 2024-03-26

**Authors:** Mick Parra, Maialen Aldabaldetrecu, Pablo Arce, Sarita Soto-Aguilera, Rodrigo Vargas, Juan Guerrero, Mario Tello, Brenda Modak

**Affiliations:** 1Laboratory of Natural Products Chemistry, Centre of Aquatic Biotechnology, Faculty of Chemistry and Biology, University of Santiago of Chile, Santiago 9160000, Chile; mick.parra@usach.cl; 2Laboratory of Bacterial Metagenomic, Centre of Aquatic Biotechnology, Faculty of Chemistry and Biology, University of Santiago of Chile, Santiago 9160000, Chile; sarita.soto@usach.cl (S.S.-A.); rvargasc@gmail.com (R.V.); 3Laboratory of Coordination Compounds and Supramolecularity, Faculty of Chemistry and Biology, University of Santiago of Chile, Santiago 9160000, Chile; maialen.aldabaldetrecu@usach.cl (M.A.); pablo.arce@usach.cl (P.A.); juan.guerrero@usach.cl (J.G.); 4Aquaculture Production Unit, Universidad of Los Lagos, Osorno 5290000, Chile

**Keywords:** Piscirickettsiosis, *Salmo salar*, copper (I) complex, immune system, intestinal microbiota, *Piscirickettsia salmonis*, antimicrobial, Atlantic salmon

## Abstract

*Piscirickettsia salmonis* is the pathogen that most affects the salmon industry in Chile. Large quantities of antibiotics have been used to control it. In search of alternatives, we have developed [Cu(NN_1_)_2_]ClO_4_ where NN_1_ = 6-((quinolin-2-ylmethylene)amino)-2H-chromen-2-one. The antibacterial capacity of [Cu(NN_1_)_2_]ClO_4_ was determined. Subsequently, the effect of the administration of [Cu(NN_1_)_2_]ClO_4_ on the growth of *S. salar*, modulation of the immune system and the intestinal microbiota was studied. Finally, the ability to protect against a challenge with *P. salmonis* was evaluated. The results obtained showed that the compound has an MIC between 15 and 33.9 μg/mL in four isolates. On the other hand, the compound did not affect the growth of the fish; however, an increase in the transcript levels of IFN-γ, IL-12, IL-1β, CD4, lysozyme and perforin was observed in fish treated with 40 μg/g of fish. Furthermore, modulation of the intestinal microbiota was observed, increasing the genera of beneficial bacteria such as *Lactobacillus* and *Bacillus* as well as potential pathogens such as *Vibrio* and *Piscirickettsia*. Finally, the treatment increased survival in fish challenged with *P. salmonis* by more than 60%. These results demonstrate that the compound is capable of protecting fish against *P. salmonis*, probably by modulating the immune system and the composition of the intestinal microbiota.

## 1. Introduction

The aquaculture industry occupies an important place in the economy of Chie, positioning the country as the second largest producer of salmonids worldwide after Norway [1]. Favorable climatic and environmental conditions have facilitated rapid growth in this productive sector in Chile, with Atlantic salmon, coho salmon and rainbow trout being the main species produced [2]. However, this explosive growth has generated an increase in outbreaks of infectious diseases, because fish are exposed to constant stress due to handling, transportation, higher density, changes in temperature and salinity, among others, where these conditions increase their susceptibility to different types of pathogens, which cause significant economic losses to the industry [3]. The pathogens that most affect the salmon industry are bacteria, mainly *Piscirickettsia salmonis*, *Tenacibaculum dicentrarchi* and *Flavobacterium psycrhophilum* [2]. In the case of *P. salmonis*, this bacterium has been the infectious agent that has caused the most deaths in salmonids for more than 30 years; it is estimated that the losses associated with this pathogen range between USD 700 and USD 800 million annually [4]. In 2022, *P. salmonis* was the cause of 50% of pathogen mortality in Atlantic salmon and rainbow trout [2]. *P. salmonis* is the etiological agent of salmonid rickettsial syndrome (SRS); it is a Gram-negative, pleomorphic but generally coccoid bacteria, with a variable size between 0.2 μm and 1.5 μm [5] and facultative intracellular [6]. *P. salmonis* is able to survive and replicate within replicative vacuoles in phagocytic cells such as macrophages and monocyte-like cells [7] by evading phagosome-lysosome fusion [8]. On the other hand, this bacterium is capable of modulating different biological processes in the host, including the cellular and humoral immune response [9,10]. Fish infected with *P. salmonis* exhibit lethargy, anorexia, darkening of the skin, swimming at the surface, and respiratory distress. It is possible to observe external lesions such as perianal and periocular hemorrhages, abdominal petechiae, superficial hemorrhagic ulcers of variable size ranging from 0.5 to 2 cm in diameter and white nodules of more than 1 cm in diameter [11]. Internally, the presence of yellow nodules in the liver is observed [12], as are generalized internal pallor, inflammation and multifocal pale areas in the kidney and spleen [11], affecting the brain, heart, skeletal muscle, gills and ovaries [5]. To control the disease caused by this bacterium, vaccines and antibiotics are mainly used. However, due to limited knowledge of the immune system in teleosts, these vaccines have not been effective [4,13]. Since vaccines are not sufficient to control this disease, the Chilean salmon industry uses a large amount of antibiotics. In 2022, 341.5 tons of antibiotics were used, of which 91.28% (approximately 304.6 tons) of the antibiotics were used against *P. salmonis* [14]. This large amount of antibiotics used has caused various problems, such as the development of *P. salmonis* isolates resistant to antibiotics [15,16]. On the other hand, environmental pollution is generated, because antibiotics are administered to the salmon through food, so it leaks from the cages into the water and marine sediment in which they are found. In addition, there is a flow of traces of antibiotics present in salmon urine and feces into the marine environment [17,18]. Another problem of excessive use of antibiotics, reported in recent years, is the effect on the composition of the intestinal microbiota of salmonids. Antibiotics generate dysbiosis in the intestine of fish, which causes changes in various biological processes, including the immune response [19,20,21]. One of the alternatives is the use of natural compounds, which, when administered in diets, causes positive effects in the fish organism. For example, a wide range of polyphenols have been studied as dietary supplements for fish, observing that they have antioxidant effects, modulate the immune response, generate protection from diseases, and improve reproduction and growth in a wide variety of fish. In addition, coumarin derivatives have attracted considerable attention due to their biological functions in fish, including antiparasitic, antibacterial and antiviral activities [22,23,24]. On the other hand, currently, there is great interest in copper coordination compounds based on ligands with the N- donor atom. This interest arises from its antiproliferative, antibacterial, antiviral, and cytotoxic properties [25], as well as its ability to modulate the immune response [26,27]. In the aquaculture industry, copper has historically been used as a treatment against different pathogens such as *Vibrio*, *Aeromonas* and *Flavobacterium*, among others, where copper (II) sulfate, CuSO_4_, is the most widely used form of copper supply. A study carried out with the marine pathogens *Aeromonas hydrophila* and *Flavobacterium columnare* showed that CuSO_4_ had an MIC of 83.2 ± 0 mg/L towards *A. hydrophila*, while the IC_50_ of CuSO_4_ towards *F. columnare* was 4.8 ± 0.3 mg/L, and the minimum bactericidal concentration of CuSO_4_ towards *F. columnare* was 25.0 ± 0 mg/L [28,29].

The adaptability of transition metal compounds, in particular, Cu(I) coordination complexes with bidentate nitrogen donor ligands, is well known. Therefore, several studies have been conducted towards the controlled obtaining of its electrochemical properties through the selection of its ligands [30]. Considering the need to use alternatives to antibiotics, in addition to the positive effects of natural compounds and their derivatives, added to the properties that transition metal complexes possess, our working group has synthesized a new [Cu(NN_1_)_2_]ClO_4_ complex, where NN_1_ is a 6-((quinolin-2-ylmethylene)amine)-2H imine ligand-chrome-2-one, a derivative of the natural compound coumarin. This complex showed a better antibacterial effect in vitro against *F. psychrophilum* than its precursors, coumarin and copper (I) salt [31], and when administering 60 μg/g of fish to rainbow trout, it generates protection against *F. psychrophilum* [32]. Considering the previous results of the effect of this compound on *F. psychrophilum*, in the present work, we analyze its antibacterial potential on *P. salmonis* and the effect of its administration in Atlantic salmon on the immune response, its modulation of the intestinal microbiota and its ability to generate protection against infection with a *P. salmonis* isolate such as LF-89. The results obtained in this work showed the different effects of the compound, such as the effects of antibacterial activity on *P. salmonis* and the ability to modulate the immune response and intestinal microbiota in Atlantic salmon. These multiple effects of the compound could be the mechanisms by which it generates protection against *P. salmonis* in Atlantic salmon. These are promising results for this compound as an alternative treatment for Piscirickettsiosis.

## 2. Results

### 2.1. Synthesis of Cu (I) Coordination Complex

Previously, in our laboratory, we have synthesized the coordination complex [Cu(NN_1_)_2_]ClO_4_, where NN_1_ = 6-((quinolin-2-ylmethylene)amino)-2H-chromen-2-one, a ligand derivate from natural product coumarin 1-benzopyran-2-one [31]. In this paper, we account for the changes made in the experimental procedure to improve the process. On this occasion, synthesis was easily achieved using the template condensation method from equimolar amounts of reagents, as shown in Figure 1, which allowed the reaction time to be reduced to 1 h, increasing the reaction yield to 90%, at room temperature and without requiring pressure systems, substantially improving the energy and economic cost of the process.

### 2.2. Antibacterial Activity of [Cu(NN_1_)_2_]ClO_4_

The antibacterial activity of [Cu(NN_1_)_2_]ClO_4_ was evaluated in four isolates of *P. salmonis*. Isolate 8149 was the most sensitive with an MIC value of 15.0 + 7.1 and an IC_50_ value of 5.3 + 0.8 μg/mL. On the other hand, isolates CGRO2, 12,201 and 727 were more resistant to the compound with MIC values between 29 and 34 μg/mL and IC_50_ values between 11 and 15 μg/mL, approximately (Table 1).

### 2.3. Effect of [Cu(NN_1_)_2_]ClO_4_ on the Growth of Atlantic Salmon

The safety of the administration of [Cu(NN_1_)_2_]ClO_4_ for 60 days in Atlantic salmon was evaluated. During feeding, no negative effects on salmonid behavior were observed. Similarly, no changes were observed in the growth of the salmonids during the 60 days of experimentation (Figure 2).

### 2.4. Evaluation of the Administration of [Cu(NN_1_)_2_]ClO_4_ on Immune Status of Atlantic Salmon

The immunostimulation capacity of natural compounds has been widely studied in various organisms, including coumarins. Because our compound, [Cu(NN_1_)_2_]ClO_4_, which has coumarin as a natural ligand, its ability to stimulate the immune system was evaluated. Fifteen days after the administration of [Cu(NN_1_)_2_]ClO_4_, in fish treated with 40 μg/g of fish of the compound, an increase in INF-γ transcript levels close to 7-fold, an increase in IL-1β close to 5-fold, an increase in CD4 close to 3-fold, an increase in perforin close to 2-fold, and a 2.4-fold increase in lysozyme was observed. On the other hand, in fish treated with 60 μg/g of fish, only an increase in IL-1β transcript levels close to three-fold was observed (Figure 3a). At 30 days of [Cu(NN_1_)_2_]ClO_4_ administration, IFN-γ transcript levels increased five-fold, while TNF-α increased close to nine-fold in fish treated with 40 μg/g of fish (Figure 3b). After 45 days of administration of [Cu(NN_1_)_2_]ClO_4_, no statistically significant changes were observed in the transcript levels of the genes under study (Figure 3c). Finally, after 60 days of [Cu(NN_1_)_2_]ClO_4_ administration, the fish treated with 20 μg/g of fish showed an increase close to four times in the INF-γ transcript levels; on the other hand, a decrease in the CD4 and lysozyme transcript levels close to 4- and 50-fold, respectively, were observed. Fish treated with 40 μg/g fish of [Cu(NN_1_)_2_]ClO_4_ showed a close to six-fold increase in IL-1β transcript levels, while a close to two-fold decrease was observed in CD4. On the other hand, fish treated with 60 μg/g of fish of [Cu(NN_1_)_2_]ClO_4_, showed a decrease in the transcript levels of CD4, lysozyme, and perforin, between 2-fold and 3-fold (Figure 3d).

### 2.5. Effect of the Administration of [Cu(NN_1_)_2_]ClO_4_ on the Composition of the Intestinal Microbiota of Atlantic Salmon

The intestinal microbiota of the experimental fish feed with [Cu(NN_1_)_2_]ClO_4_ showed a different pattern to the intestinal microbiota of the control group. In control fish, the intestinal microbiota was composed of bacteria belonging to the Phylum Proteobacteria (97.8%), followed by Actinobacteria (0.65%), Firmicutes (0.38%) and Bacteroidetes (0.37%). At the phylum level, the administration of the compound [Cu(NN_1_)_2_]ClO_4_ increased the proportion of Proteobacteria to 98.9% and the presence of an ASV of bacterial origin that could not be classified at the Phylum level to 0.22%, while the phylum Firmicutes (0.3%) and Bacteroidetes (0.22%) showed a slight decrease in their percentage of relative abundance (Figure 4). At the genus level, 24 different genera were found to be present in both groups of fish, while 32 genera were present only in control fish, and 53 only in the treated fish. When analyzing the bacterial genera with an abundance greater than 0.05%, we found that 20 different genera were present in both groups, 24 only in the control group and 8 only in the treated group. Furthermore, administration of the compound decreased the relative abundance of 14 different bacterial genera, while it increased the relative abundance of 5 bacterial genera. The bacterial genus with the highest abundance in control and treated fish was *Photobacterium* with 95.86% and 97.32%, respectively. In fish treated with [Cu(NN_1_)_2_]ClO_4_, it was possible to find bacterial genera that were not found in the control fish, such as *Lactococcus* (0.09%), *Lactobacillus* (0.05%) and *Bacillus* (0.05%), that have the potential to be used as bacterial probiotics. On the other hand, it was also possible to find pathogenic bacteria of the genus *Piscirickettsia* (0.11%) (Figure 5).

### 2.6. Evaluation of the Protective Effect of [Cu(NN_1_)_2_]ClO_4_ against a Challenge with P. salmonis

The effect of administering 40 μg/g of [Cu(NN_1_)_2_]ClO_4_ to fish during a challenge with *P. salmonis* was evaluated. The results showed that the compound administered in the feed generated protection in Atlantic salmon when challenged with *P. salmonis*. A decrease in mortality could be observed in the fish treated with the compound, with 90% of the treated fish surviving, while only 30% of the untreated fish survived during 30 days of experimentation. The survival analysis was carried out using the Kaplan–Meier estimate, performing a log-rank test (Mantel–Cox), obtaining a statistically significant difference with a value of *p* < 0.0001 (Figure 6a). The analysis of bacterial loads in dead fish from both treated and untreated fish showed similar values, between 1 × 10^2^ and 1 × 10^4^ the number of copies/50 ng of sample. On the other hand, in the surviving fish, those treated with [Cu(NN_1_)_2_]ClO_4_, it was not possible to detect the gene of *P. salmonis* in any of the fish, while in those without treatment, two of the fish had a bacterial load close to 1 × 10^2^ copy number/50 ng of sample (Figure 6b). These results show that the administration of [Cu(NN_1_)_2_]ClO_4_ in Atlantic salmon reduces the mortality generated by *P. salmonis* and decreases the amount of pathogen in the fish organism.

## 3. Discussion

The aquaculture industry in Chile is one of the most important economic sectors. However, the rapid growth of this productive sector has generated the continuous appearance of pathogens that affect the different production species. To face this problem, the Chilean aquaculture industry has used a large amount of antibiotics, generating various environmental problems and devaluing the Chilean product. In search of new alternatives, our working group synthesized the Cu (I) complex [Cu(NN_1_)_2_]ClO_4_ using coumarin as a ligand, achieving improvements in the antibacterial capacity against *F. psycrhophilum* compared to its precursor molecules (coumarin and [Cu(CH_3_CN)_4_]ClO_4_) [31] and generating protection for rainbow trout against a challenge with *F. psycrhophilum* [32]. Considering these previous works, in this article, we evaluate the antibacterial capacity of this compound against *P. salmonis*. As well as its ability to stimulate the immune response in Atlantic salmon and generate protection against a challenge with *P. salmonis*. The importance of working with this bacterium is because it is the most relevant in the Chilean salmon industry, being responsible for the highest mortalities in the industry for more than 30 years.

The results obtained showed that the compound is capable of reducing the growth of different isolates of *P. salmonis*, in concentrations similar to what was previously reported against *F. psychrophilum* [31]. On the other hand, no considerable differences were observed in the antibacterial activity of the compound between the *P. salmonis*-like LF-89 and *P. salmonis*-like EM-90 isolates. Although only four isolates were used in this work, it is relevant to identify compounds that have a similar effect between different isolates and genogroups of *P. salmonis*. Genotypical and phenotypical differences at the level of isolates and genogroups have been widely studied in *P. salmonis* [33]. These differences affect the sensitivity of the isolates to different antibiotics, making it even more difficult to find a treatment effective against several *P. salmonis* outbreaks [16,34]. Considering these results, the [Cu(NN_1_)_2_]ClO_4_ complex is a promising compound for a common treatment effective against a wide spectrum of isolates from different genogroups; however, it is necessary to perform additional experiments using a wider range of *P. salmonis* isolates to confirm this property.

The administration of the compound through feeding for 60 days did not appear to affect the growth of Atlantic salmon. These results are similar to those previously reported for this compound in rainbow trout [32]. On the other hand, the same effects have been observed in other nutritional supplements for salmonids based on natural products [35,36].

Although the compound was initially designed as an antibacterial, in this work, we also studied the capacity of [Cu(NN_1_)_2_]ClO_4_ to stimulate the cellular immune response in Atlantic salmon and its effect on the composition of the intestine microbiota. The administration of [Cu(NN_1_)_2_]ClO_4_ modulated the transcription levels of all the immune system markers analyzed in their different concentrations and times tested. [Cu(NN_1_)_2_]ClO_4_ at a concentration of 40 μg/g of fish generated the most important immunostimulating effect in this work, increasing the levels of lysozyme, perforin, IFN-γ, TNF-α, IL-1β and CD4. In the case of perforin, a glycoprotein that forms pores in the cell membrane of target cells [37], and which, in mammals, is expressed in NK cells, CD8+ T lymphocytes and CD4+ T lymphocytes, its antibacterial and antiviral in fish widely studied. For example, in Atlantic salmon, an increase in perforin is observed in the inhibition of Rock bream iridovirus (RBIV) replication [38]. Similar results were observed in Atlantic salmon against Pilchard orthomyxovirus (POMV) [39] and rohu and common carp against *Aeromonas hydrophila* [40,41]. On the other hand, the increase in lysozyme expression levels observed by treatment with [Cu(NN_1_)_2_]ClO_4_ is also related to an antibacterial effect. This enzyme hydrolyzes the peptidoglycan layer of the bacterial cell wall by cleaving the beta-1,4 glycosidic bonds between N-acetylmuramic acid and N-acetylglucosamine [42,43]. Lysozyme is the first line of defense against different types of pathogens such as fungi, parasites, bacteria and viruses [43,44,45]. Several studies show that feed supplementation with different secondary metabolites such as tea polyphenols increases lysozyme activity in common carp [46] and Asian sea bass [47], as well as flavonoids in northern snakehead fish [48]. In the case of cytokine markers, administration of [Cu(NN_1_)_2_]ClO_4_ markedly increased INF-γ transcript levels. The functions of IFN-γ are several, among which the activation of macrophages and CD4+ T helper lymphocytes has been studied, a fundamental process for eliminating intracellular pathogens, in addition to promoting inflammation and the presentation of antigens, a process for the inhibition of viral replication [49,50]. On the other hand, IFN-γ activates macrophages towards the “M1” phenotype, which increases the expression of pro-inflammatory cytokines such as IL-12, IL-1β and TNF-α, in addition to reactive oxygen species [51]. Moreover, IL-12, IL-1β and IL-8 promote the synthesis of IFN-γ, while TGF-β represses the expression of IFN-γ [52]. Various cell types such as Natural Killer (NK), CD4+ T helper (Th) cells, and cytotoxic CD8+ cells, among others, produce IFN-γ [52,53]. The transcript levels of IL-1β, TNF-α and CD4 were also increased in fish treated with [Cu(NN_1_)_2_]ClO_4_, demonstrating a relationship in the ability to stimulate the cellular immune response of this compound. Various studies, in which different molecules are administered in fish, have demonstrated a relationship between the increase in the transcription levels of pro-inflammatory and anti-inflammatory cytokines and protection against different pathogens. For example, caffeic acid and ferulic acid increase the transcript levels of IFN-γ, IL-1β and TNF-α in Nile Tilapia, generating protection against *A. veronii* [54,55]. Trans-cinnamic acid increases IFN-γ, IL-1β, IL-8, TGF-β and TNF-α, increasing the survival of rainbow trout against *Y. ruckeri* [56]. Coumarin increases IFN-γ levels, generating protection in zebrafish infected with spring viraemia of carp virus (SVCV) [23,57]. The immunomodulatory effect of the compound could be the mechanism of action through which the compound is capable of generating protection against *P. salmonis*. Similar mechanisms of action were reported for treatments based on natural compounds against *P. salmonis* [35,36].

[Cu(NN_1_)_2_]ClO_4_ also showed the ability to modify the intestinal microbiota composition of Atlantic salmon. The fish used in this study had a microbiota composed mainly of Proteobacteria, particularly dominated by Photobacterium. This composition deviates somewhat from the commonly observed fish microbiota, which is typically dominated by Firmicutes and Proteobacteria [58]. This discrepancy could be attributed to the cultivation conditions favoring the presence of *Photobacterium* in the Atlantic salmon specimens used. High relative abundance values have also been reported in freshwater Atlantic salmon specimens, with values of 40 ± 25% [59]. On the other hand, the administration of [Cu(NN_1_)_2_]ClO_4_ generated a decrease in the abundance of bacteria of the *Corynebacterium* genus, associated with opportunistic pathogens that strongly affect the salmon industry. In addition, it was observed that it modulates the abundance of bacteria of the genus *Vibrio*, in which both pathogenic bacteria and bacteria with probiotic potential can be found [60]. Moreover, various bacterial genera widely studied for their probiotic potential, such as *Pseudomonas*, *Lactobacillus*, *Lactoccocus* and *Bacillus*, increased their abundance [61,62,63]. This effect of modulating the composition of the intestinal microbiota, through the incorporation of different nutrients in the fish diet, has been widely studied. For example, administration of the prebiotic Selectovit in Atlantic salmon increases the abundance of *Bacillus* and *Mycoplasma* [64]. In turbot, the administration of daidzein has a dual effect, increasing the abundance of lactic acid bacteria but also potential pathogens such as *Prevotella copri* [65]. Similar results were reported in tilapia fed with resveratrol, modulating the taxa of potential pathogenic and commensal bacteria, depending on the concentration [66]. Finally, it was observed that the administration of the compound surprisingly increased the abundance of bacteria of the genus *Piscirickettsia*, to which *P. salmonis* belongs. However, it has been reported that this bacterium is part of the intestinal microbiota of healthy fish, and that an imbalance in the composition of the microbiota and the interaction between different metabolites of various bacteria generates the appearance of SRS [67]. These data are relevant when interpreting our results, because although the compound generated an increase in this bacterial genus, the administration of [Cu(NN_1_)_2_]ClO_4_ during the challenge with *P. salmonis* generated protection, considerably increasing the survival of Atlantic salmon. This suggests that the compound generates a balance in the intestinal microbiota at the time of challenge. Finally, considering that in vitro the compound has an antibacterial effect, and that in vivo it can modulate the immune response and the composition of the intestinal microbiota. The protective effect observed may be multifactorial, attacking different fronts to increase the survival of Atlantic salmon against *P. salmonis*.

## 4. Materials and Methods

### 4.1. Synthesis of Cu (I) Coordination Complex

The synthesis of Cu(I) coordination compound [Cu(NN_1_)_2_]ClO_4_, where NN_1_ is the ligand 6-((quinolin-2-ylmethylene)amino)-2H-chromen-2-one was carried out using the template condensation method [30]. To 3 mmol of the precursor reagent [Cu(CH_3_CN)_4_]ClO_4_ dissolved in 50 mL of acetonitrile, double the amounts of the reactants 2-quinoline-carboxaldehyde (6 mmol) and 6-amino-chromen-2-one (6 mmol), dissolved in acetonitrile, were added and maintained at room temperature and constant stirring for 1 h, forming a colored solution. The volume of solution was reduced in a rotary evaporator and the concentrate was precipitated with cold ethyl ether. The microcrystalline precipitate was recrystallized from an ethyl ether/acetonitrile (9:1) mixture and finally washed with ethyl ether. The chemical structure was confirmed by comparing the signals obtained from the NMR^1^H spectrum of the complex obtained, with what was previously reported [31].

### 4.2. Fish and Maintenance

Pre-smolt Atlantic salmon (*S. salar*) weighing between 25 and 35 g (Blumar, Los Angeles, Chile) was used. One week before the start of the experiments, the fish were acclimatized in ponds with a biomass of 14 g/L and a temperature of 12 °C and fed daily at 1% of body weight (Golden Optima, Biomar, Chiloé, Chile). The water parameters were monitored daily, maintaining the pH at between 6.6 and 7, while the salinity was adjusted to 6 PSU. On the other hand, total ammonia was monitored to maintain values below 0.02 mg/L. Each day, changing the water in the ponds, measuring pH, salinity, and ammonia, and feeding the fish were performed manually. The experiments were carried out in accordance with the ethical standards of the Institutional Ethics Committee of the University of Santiago de Chile and the relevant current legislation. The authorization from the Ethics Committee of the University of Santiago de Chile to carry out the experiments with fish in the FONDEF VIU project was approved with number 354.

### 4.3. Antibacterial Activity

To determine the MIC and IC_50_ of [Cu(NN_1_)_2_]ClO_4_ on *P. salmonis*, the microdilution method was used [68] with some modifications. For this experiment, four isolates of *P. salmonis* were used (CGRO2, 12,201 and 8149 provided by Dr. Veronica Cambiazo, bioinformatics and gene expression laboratory, INTA, University of Chile and 727 provided by Dr. Aldo Gaggero, virology laboratory, ICBM, Universidad de Chile), which were grown in cell-free medium (Austral-SRS medium) [6]. The bacteria were incubated on plates with Austral-SRS agar medium for 5 days at 18 °C. Subsequently, the bacteria were collected and inoculated in 5 mL of Austral-SRS medium and incubated for 3 days at 18 °C with shaking at 180 rpm. Bacteria were adjusted to OD_600_ = 0.1 and inoculated again into 10% Austral-SRS medium in a final volume of 5 mL and incubated for 3 days at 18 °C with shaking at 180 rpm. Subsequently, *P. salmonis* isolates were inoculated at an OD_600_ = 0.1 in a 10% cell suspension in AUSTRAL-SRS medium in 96-well plates (SPL), and treated with [Cu(NN_1_)_2_]ClO_4_ in serial concentrations between 128 µg/mL and 2 µg/mL. The plates were incubated for 96 h at 16 °C with shaking at 180 rpm. To calculate the MIC and IC_50_ of [Cu(NN_1_)_2_]ClO_4_ in the four *P. salmonis* isolates, the OD_600_ nm of each well was measured using Nanoquant Infinite M200 Pro (TECAN, Grödig, Austria). Data were analyzed using GraphPad Prism 8.0 software. Concentrations were log (10) transformed, OD_600_ nm was normalized as a percentage, and nonlinear regression was performed to calculate MIC and IC_50_.

### 4.4. Evaluation of the Immune Response and Modulation of the Intestinal Microbiota

The evaluation of the immune system and the composition of the intestinal microbiota was carried out using 120 Atlantic salmon of 25 ± 2 g, which were divided into 5 groups, each group with 24 fish divided into 2 aquariums. Group A contained untreated fish (Ctrl), group B fish were treated with 20 µg/g of [Cu(NN_1_)_2_]ClO_4_ (20), group C fish were treated with 40 µg/g of [Cu(NN_1_)_2_]ClO_4_ (40), and group D fish were treated with 60 µg/g of [Cu(NN_1_)_2_]ClO_4_ (60). All the fish were fed for 60 days with commercial pellets plus the respective treatments, mixed with commercial oil, while the control fish were only fed with the commercial pellet plus oil. Three fish were sacrificed per aquarium at 15, 30, 45, and 60 days of experimentation (*n* = 6 per treatment); then, the fish were weighed and the head kidney, and intestine were removed. The samples were stored at −80 °C.

### 4.5. Growth of P. salmonis in Cell Culture

The challenge experiment was performed with *P. salmonis* isolate 727 (like LF-89). For the growth of this bacteria, the SHK-1 cell line (ECACC) was used, which was grown in L-15 medium (Cytiva, Hyclone, South Logan, UT, USA) supplemented with 10% fetal bovine serum (Cytiva, Hyclone, South Logan, UT, USA), 4 mM L-glutamine (Mediatech, Corning, Manassas, VA, USA) and 40 μM β-mercaptoethanol (Life technologies, Gibco, New York, NY, USA), in T175 cell culture bottles (SPL). The cells at 80% confluence were incubated with *P. salmonis* for 24 h; later, the cells were washed twice with PBS 1X (Cytiva, Hyclone, South Logan, UT, USA) and incubated for 2 h with 50 μg/mL gentamicin. Subsequently, the cells were washed twice with PBS 1X and incubated with fresh L-15 medium, described previously, for 12 days at 16 °C. The cells were collected using a cell scraper and centrifuged at 1000× *g* to precipitate the cells, the supernatant was collected and centrifuged at 5000× *g* to collect the bacteria, the supernatant was discarded, and the pellet containing the bacteria was resuspended in 1 mL of physiological serum.

The quantification of *P. salmonis* was performed using the kit LIVE/DEAD BacLight bacterial viability and counting kit (Life technologies, Carlsbad, CA, USA) according to the manufacturer’s instructions.

### 4.6. Evaluation of the Protective Effect of [Cu(NN_1_)_2_]ClO_4_

The ability of [Cu(NN_1_)_2_]ClO_4_ to protect Atlantic salmon against *P. salmonis* was evaluated. For this experiment, 60 Atlantic salmon 35 ± 2 g were used, divided into 3 different groups with 10 fish per pond in duplicate (total *n* = 20). Group A: Fish fed with pellets mixed with commercial oil (untreated). Group B: Fish fed with pellet mixed with commercial oil (challenge control). Group C: Fish fed with [Cu(NN_1_)_2_]ClO_4_ at 40 µg/g of fish, mixed in the pellet with the commercial oil. All fish were fed with their respective treatments for 15 days before being challenged with 1 × 10^7^
*P. salmonis* bacteria in 100 µL, via intraperitoneal injection (previously anesthetized with 30 mg/L benzocaine) in the case of group B and C, while group A was injected with 100 µL of physiological serum. Subsequently, all fish were fed throughout the experiment with their respective treatments. Daily mortality was recorded for 30 days after challenge.

### 4.7. RNA Extraction

To extract the RNA from the kidneys of the fish, the TRIzol reagent (Invitrogen, Carlsbad, CA, USA) was used, following the manufacturer’s instructions, with some modifications (Ref parra 2020). The RT reaction was performed with the All-In-One 5X RT MasterMix kit (ABM, Richmond, BC, Canada), using 2 µg of RNA, 4 µL of the master mix and nuclease-free H_2_O to a final volume of 20 µL. The thermal profile used was 30 min at 37 °C, 10 min at 60 °C and 3 min at 95 °C.

### 4.8. DNA Extraction

DNA extraction from the intestine and anterior kidney was performed using Wizard^®^ Genomic DNA Purification Kit (Promega, Madison, WI, USA) according to the manufacturer’s instructions.

### 4.9. Real-Time PCR

The quantification of the transcript levels of the immunological genes was carried out using real-time PCR. Reactions were performed in 96-well plates (Thermoscientific, Waltham, MA, USA) using the PikoReal 96 Real-Time PCR System (Thermoscientific). The reaction mixture consisted of 5 μL of SsoAdvanced Universal^™^ SYBR^®^ Green Supermix (Biorad, South Granville, Australia), 0.5 μL of each primer (10 uM), 1 μL of cDNA (80 ng) and 3 μL of ultrapure water (Invitrogen) to complete 10 μL. Subsequently, the transcript levels of the genes (IL-12, IFN-γ, IL-1β, TNF-α, TGF-β, CD4, Perforin and Lysozyme) were quantified. The thermal profile used was 1 cycle at 95 °C for 2 min and 40 cycles at 95 °C for 5 s, 60 °C for 15 s, and 72 °C for 15 s. To analyze the change in the level of transcripts of each gene under study, elongation factor 1α (ef1a) was used to normalize the expression of target genes using the ΔΔCT method. [69]. Statistically significant differences were determined by comparing the control with each treatment, using a one-way nonparametric *t*-test (Mann–Whitney) (* *p* < 0.05, ** *p* < 0.01, *** *p* < 0.001).

The quantification of the bacterial load was carried out by detecting the 16S rRNA gene of *P. salmonis*. The reaction was carried out according to the aforementioned protocol, adding 50 ng of DNA. The thermal profile used was 1 cycle at 95 °C for 2 min, 35 cycles at 95 °C for 5 s, 60 °C for 15 s, and 72 °C for 15 s. To calculate the number of copies of the gene, a previously made calibration curve was used. The primers used in these experiments are listed in Appendix A.

### 4.10. 16 S Ribosomal Sequencing

The 16 S rRNA amplification and high-throughput sequencing was performed by Molecular Research LP (MR DNA; Shallowater, TX, USA) following the protocol described by Vargas et al. 2023 [70]. Briefly, each sample of DNA was diluted to 25 ng/μL. The DNA of three fish per pond were mixed in equal volumes to form a pool. For each condition, two pools were generated. Each DNA pool was used as a template to amplify the V4 variable region using primers 515-F and 806-R [71] with a HotStarTaq Plus Master Mix Kit (Qiagen, Hilden, Germany). The PCR conditions were set up to 94 °C (3 min), 30 cycles of 94 °C (30 s), 53 °C (40 s), and 72 °C (60 s), with a final elongation to 72 °C (5 min). PCR products were analyzed in a 2% agarose gel, purified using Ampure XP beads. Illumina DNA library using purified PCR products was prepared with a TruSeq Nano kit. MiSeq reagent kit v3 on the Illumina MiSeq platform (2 × 300-bp paired ends [PE]) was used for high-throughput sequencing of amplicons following the manufacturer’s guidelines.

### 4.11. Bioinformatic Analysis

The output data from sequencing were processed following the pipeline described by Vargas et at 2023 [70]. Briefly, data were processed using QIIME2 version 2020.2 [72]. DADA2 [73] was used for quality filtering and prediction of Amplicon Sequence Variants (ASV). The reads (forward and reverse) were truncated to 250 nts. ASVs/features were taxonomically classified using a pre-trained Naive Bayes taxonomy classifier, Greengenes 13,899% ASVs [74]. We obtained a mean of 108,898 individual sequencing reads per sample (min = 5166; max = 55,565). After data processing, the average number of sequences for each sample passing through ASV classification was 185276 (SD: 105,416). The average number of ASVs per sample was 30,381 (SD: 19,314). Diversity analyses were performed using the same number of random reads obtained by rarefaction with a sampling depth of 26.200 per sample. The very low-abundance taxa (<0.05%) and taxa not represented in at least half the samples were removed for further analysis.

## 5. Conclusions

[Cu(NN_1_)_2_]ClO_4_, a copper (I) complex developed in our laboratory, is capable of inhibiting the growth of *P. salmonis* isolates like LF-89 and EM-90, demonstrating its broad effect on this bacterium. On the other hand, the administration of the compound through diet is capable of modulating both the intestinal microbiota and the cellular immune response in Atlantic salmon; however, we cannot determine whether the effect of the compound modulates the microbiota, which in turn modulates the immune response, or whether they are independent effects. Future work is needed to determine this. On the other hand, these could be the mechanisms of action by which the compound is capable of generating protection against a challenge against *P. salmonis*. Finally, the results obtained in this work demonstrated the potential of using [Cu(NN_1_)_2_]ClO_4_ as a treatment against *P. salmonis*, an intracellular pathogen of great relevance in the Chilean salmon industry, representing a promising alternative to the use of antibiotics. However, it is necessary to continue carrying out tests and studies to increase the characterization of the effect of the compound, such as the evaluation of toxicity on other aquatic organisms, its accumulation on the seabed, and synergism effect with antibiotics, among other experiments.

## Figures and Tables

**Figure 1 ijms-25-03700-f001:**
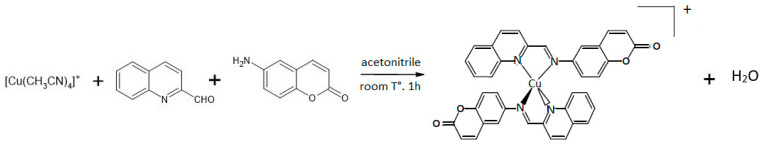
Synthesis scheme of the complex [Cu(NN_1_)_2_]ClO_4_ using the template method.

**Figure 2 ijms-25-03700-f002:**
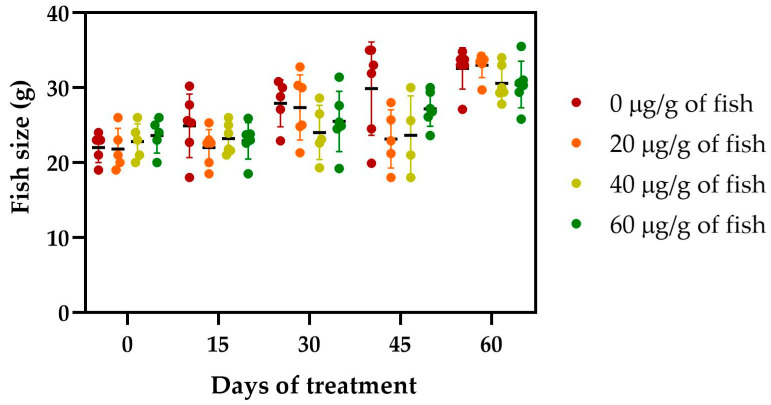
Effect of the [Cu(NN_1_)_2_]ClO_4_ on the growth of Atlantic salmon during the 60 days of experimentation. Every 15 days, 6 fish per condition were sampled and weighted. The legend indicates the doses tested, 0 µg/g of fish (Control), 20 µg of [Cu(NN_1_)_2_]ClO_4_/g of fish of, 40 µg of [Cu(NN_1_)_2_]ClO_4_/g of fish, and 60 µg of [Cu(NN_1_)_2_]ClO_4_/g of fish. The significance was analyzed using the Mann–Whitney test.

**Figure 3 ijms-25-03700-f003:**
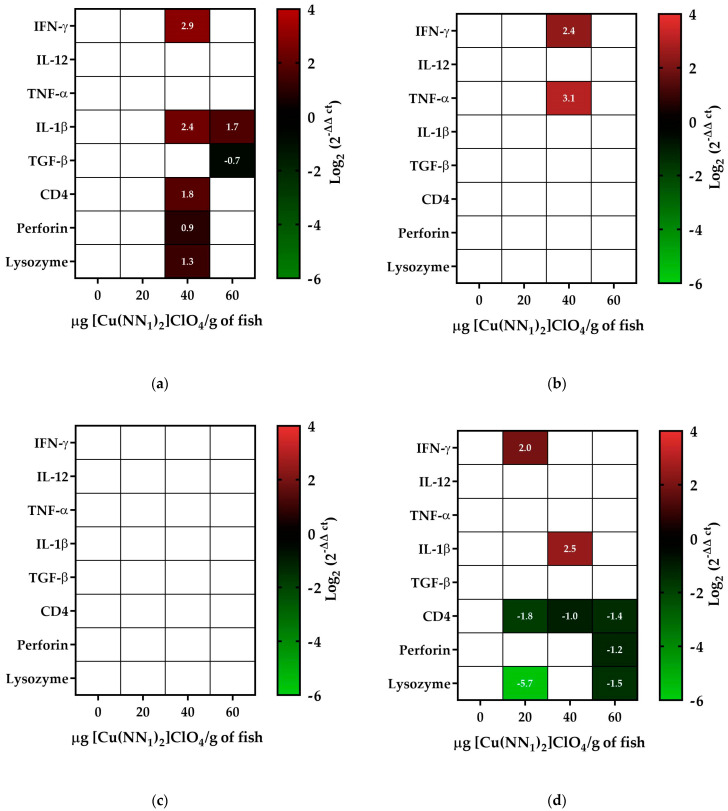
Effect of oral administration of [Cu(NN_1_)_2_]ClO_4_ on the immune system of Atlantic salmon. The figure shows the relative transcript levels of genes encoding for Interferon-γ (IFNγ), Interleukin 12 (IL-12), Tumon Necrotic Factor α(TNF-α), Interleukin 1β (IL-1β), Tumon Growth Factor-β (TGF-β), CD4, Perforin and lysozyme. The fish were treated with 0 (Ctrl), 20 µg, 40 µg and 60 µg of [Cu(NN_1_)_2_]ClO_4_ per gram of fish during 15 (**a**), 30 (**b**), 45 (**c**), and 60 (**d**) days of administration. The level of expression was determined by RT-qPCR, normalized with respect to the expression housekeeping gene (EF1-α). The statistically significant difference was analyzed using the Mann–Whitney test (<0.05), comparing the control with respect to the treatments. Colors represent significant difference; white cells represent non-significant difference.

**Figure 4 ijms-25-03700-f004:**
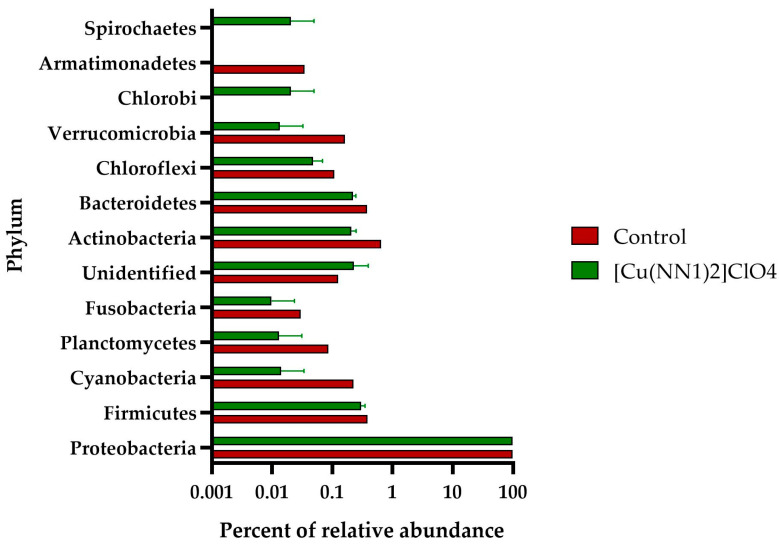
Relative abundance of Phyla identified in the gut of Atlantic salmon. The figure shows the relative abundance of the Phylum identified in the intestine of Atlantic salmon specimens from fish fed with a control diet and treated with [Cu(NN_1_)_2_]ClO_4_.

**Figure 5 ijms-25-03700-f005:**
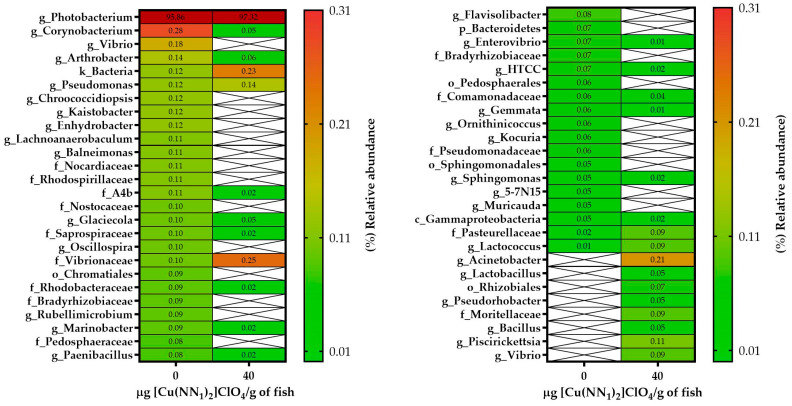
Relative abundance of genera identified in the gut of Atlantic salmon. The figure shows the relative abundance greater than 0.05% of the genera identified in the intestine of Atlantic salmon specimens from fish fed with a control diet and treated with [Cu(NN_1_)_2_]ClO_4_ (40 μg/g of fish) during 15 days. White cells with crosses mean a relative abundance equal to 0.

**Figure 6 ijms-25-03700-f006:**
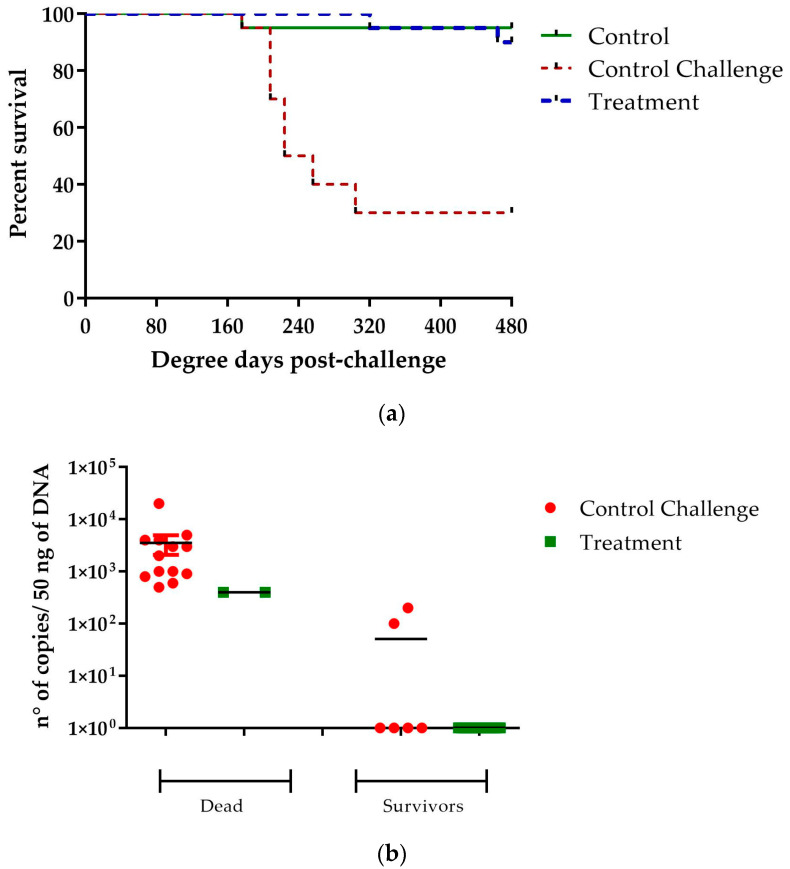
Challenge assay with *P. salmonis*. The figure shows the effect of [Cu(NN_1_)_2_]ClO_4_ on the survival (**a**) and bacterial load (**b**) of fish challenge with *P. salmonis*. The panel (**a**) shows the percentage of survival after an intraperitoneal injection with *P. salmonis* of fish feed with normal food (control) and fish feed with food supplemented with [Cu(NN_1_)_2_]ClO_4_ at 40 μg/g of fish. Panel (**b**) shows the bacterial load present in the survival and dead fish of the group control and treated with [Cu(NN_1_)_2_]ClO_4_. Statistically significant differences were determined by comparing the curves of treatment versus challenge control, using a long-rank test (Mantel–Cox) and Gehan–Breslow–Wilcoxon test (*p* < 0.0001).

**Table 1 ijms-25-03700-t001:** MIC and IC_50_ of [Cu(NN_1_)_2_]ClO_4_ on four *P. salmonis* isolates (*n* = 3).

Isolated	MIC µg/mL	IC_50_ µg/mL
CGRO2	29.0 ± 2.5	14.6 ± 5.5
12201	33.7 ± 2.4	12.6 ± 1.3
8148	15.0 ± 7.1	5.3 ± 0.8
727	33.9 ± 14.5	11.0 ± 2.0

## Data Availability

Data is contained within the article and Appendix A.

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
