# Peer review of "[Cu(NN1)2]ClO4, a Copper (I) Complex as an Antimicrobial Agent for the Treatment of Piscirickettsiosis in Atlantic Salmon"

_ijms, 2024, doi:10.3390/ijms25073700_

Round 1

Reviewer 1 Report

Comments and Suggestions for Authors

Authors brought intersting results of antibacterial activity of [Cu(NN1)2]ClO4 on Piscirickettsia salmonis and its protective activity against its infection of Salmo salar.

Unfortunately, although it is stated so in the title and in the conclusion, the substance cannot be considered antimicrobial on the basis of the results, nor can it be anticipated its possible use in the treatment of piscirickettsiosis.

The discussed compound have been previously published by same authors (Molecules 2020 25 (14)) and has shown simillar effect on Flavobacterium and its effect on rainbow trout (Microorganisms 2022 10 (11)) The antimicrobial activity is rather low (MIC for antibiotic standard is missing) and the effect is probably due to intestinal microflora changes and immunomodulation.

Tested compounds may be much more toxic and environment pollutants compared to antibiotics used in salmon production - no cellular cytotoxicity assays, no toxicity effects on other aquatic organisms (algae, protozoa, plancton, coral, fish reproduction etc), no histopathological examination of treated salmon (searching for liver, kidney, brain and gills alterations) were done. Copper content in salmon meat and skin was not examined as well. Without knowing these information, it is too bold to say that compound may be " antimicrobial agent for tereatment of piscirickettsiosis".

Source values for figures 2, 3, 4, 5 and 6, have to be added to Supplementary materials (tables with values for all samples).

In my opinion, the article is not suitable for publishing in IJMS journal, because it doesn´t meet the scope of journal (chemistry have been already published, molecular mechanisms are not known, effect is rather non-specific) After revision, the article should be resubmitted to some more suitable journal (eg. Fishes).

Author Response

Authors brought interesting results of antibacterial activity of [Cu(NN1)2]ClO4 on Piscirickettsia salmonis and its protective activity against its infection of Salmo salar.

  1. Unfortunately, although it is stated so in the title and the conclusion, the substance cannot be considered antimicrobial based on the results, nor can it be anticipated its possible use in the treatment of piscirickettsiosis.

Response: The authors appreciate the reviewer's comments, however, we do not agree, since the results demonstrated the ability of the compound to inhibit the growth of four P. salmonis isolates under in vitro conditions. In addition, the work also demonstrates the compound's ability to protect salmon against a challenge with P. salmonis. Where it is shown that in the fish that survived the challenge and fed with the compound, there is no bacterial load. Therefore, the authors consider that the molecule meets the requirements to attribute the ability to be antimicrobial.

2. The discussed compound have been previously published by same authors (Molecules 2020 25 (14)) and has shown simillar effect on Flavobacterium and its effect on rainbow trout (Microorganisms 2022 10 (11)). The antimicrobial activity is rather low (MIC for antibiotic standard is missing) and the effect is probably due to intestinal microflora changes and immunomodulation.

Response: We appreciate the comments, however, the authors do not agree with the difference mentioned by the reviewer concerning antibiotics. In this work, 40 mg/g of fish cooper complex was administered against P. salmonis, while florfenicol, used in the industry to treat P. salmonis, was administered at a concentration of 10 mg/g of fish. On the other hand, against F. pyschrophilum, previously published (Microorganisms 2022, 10(11), 2296), 60 mg/g of fish of complex were administered, while oxytetracycline, used in the industry to treat F. pyschrophilum, is administered at a concentration of 75 mg/g of fish. This shows that the administration of this compound is not far from the doses used in the salmon industry. On the other hand, in this work, we show the different effects of the complex, such as antibacterial, intestinal microflora changes, and immunomodulation. We show that it is capable of generating multiple effects on the organisms, this is normal for any treatment; no treatment has a single effect. Different works have shown that antibiotics, probiotics, and molecules administered have multiple effects. The authors do not consider it negative to demonstrate the varied effects that the molecules have, on the contrary, we consider that it enriches the knowledge and characterization of the designed compound.

3. Tested compounds may be much more toxic and environmental pollutants compared to antibiotics used in salmon production - no cellular cytotoxicity assays, nor with the characterization of the effects it has effects on other aquatic organisms (algae, protozoa, plankton, coral, fish reproduction, etc), no histopathological examination of treated salmon (searching for liver, kidney, brain and gills alterations) were done. Copper content in salmon meat and skin was not examined as well. Without knowing this information, it is too bold to say that the compound may be an "antimicrobial agent for the treatment of piscirickettsiosis".

Response: The authors are in total agreement when the reviewer says that it is necessary to carry out more studies and tests to continue advancing with the characterization of the effects it has, such as toxicity on other aquatic organisms and the seabed, to advance the commercial use of the copper complex. However, we do not consider that this means a negative point for this work; on the contrary, from the results obtained, we demonstrate that the cooper complex has the potential to continue evaluating its effect. On the other hand, remember that the accumulation of copper present in the liver, intestine, and muscle of rainbow trout was determined and published in our previous paper (Microorganisms 2022, 10(11), 2296).

Reviewer 2 Report

Comments and Suggestions for Authors

Author Response

Overall comments: The authors have developed a [Cu(NN1)2]ClO4 treatment for protection of salmon against P. salmonis infection and prevent of SRS. The treatment was shown to have a strong MIC in-vitro and robust protection in a challenge assay with live salmon exposed with P. salmonis. The treatent displayed no effect on the growth of salmon and moderate effect on the composition of the microbiota and immune response.

  • Are they any current Cu based products used to treat or prevent aquatic pathogens? If so, please cite them and state their MICs if any papers have been published with them.

Response: Copper is widely used in the aquaculture industry to prevent the growth of parasites, fungi, and bacteria. The following phrase was added to the paper: “In the aquaculture industry, copper has historically been used as a treatment against different pathogens such as Vibrio, Aeromonas and Flavobacterium, among others. Also, copper (II) sulfate, CuSO4, is the most widely used form of copper supply. Also, copper is used to control the algae and avoid fouling of fish cage netting. In the aquaculture industry, copper has historically been used as a treatment against different pathogens such as Vibrio, Aeromonas and Flavobacterium, among others, where copper (II) sulfate, CuSO4, is the most widely used form of copper supply. A study carried out with the marine pathogens Aeromonas hydrophila and Flavobacterium columnare, showed that CuSO4 had a MIC of 83.2±0 mg/L towards A. hydrophila, while the IC50 of CuSO4 towards F. columnare was 4.8±0.3 mg/L, and the minimum bactericidal concentration of CuSO4 towards F. columnare were 25.0±0 mg/L”.

  • In the “ Fish and Maintenance” section, please state the approximate age of the fish used for the study if this information is known. Could age affect the fish’s response to treatment or infection?

Response: The information on the age of the fish is not known, and estimating it is a bit complicated. Generally, we work with information on the weight of the fish for the different experiments. On the other hand, there are indeed differences in the physiological response of fish to different treatments depending on the growth stage. Adult fish have a more developed immune system, so their response to an infection is different from that of a younger fish.  Therefore, in the case of our compound, it has been evaluated in trout and salmon under laboratory conditions and in field conditions always in a weight range between 10 and 50 grams.

  • In the protective effect/challenge assay, it was said that isolate P. salmonis 727 was used. Please state why a specific strain was used? Is there known differences between the virulence of these 4 isolates?

Response: For the challenge experiments, P. salmonis isolate 727 was chosen, as it is the most studied and used isolate for this type of experiment. With this isolate, mortalities greater than 50% have been achieved in different experiments. Therefore, using this isolate assured us of having sufficient mortalities to be able to verify the protective effect of our compound under study. Although the isolates presented differences in their virulence as observed in cell culture. However, all isolates are capable of generating the cytopathic effect characteristic of P. salmonis and the loss of cell monolayer, which is important

Reviewer 3 Report

Comments and Suggestions for Authors

The work of B. Modak and co-authors studies the antibacterial activity of a copper(I) complex with coumarin derived NN’-chelating ligand for the treatment of piscirickettsiosis in Atlantic salmon. The authors demonstrated that the complex has an antimicrobial effect, and it has no effect on growth of the fish. Treatment of fish increases survival compared to controls. The compound can be considered as a promising alternative to antibiotics in the aquaculture industry.

The referee, as a specialist in chemistry, addresses the following questions to the chemical part of the work:

The abstract contains only the abbreviated formula of the copper complex - [Cu(NN1)2](ClO4), it is necessary to give the decoding of the ligand when first mentioned - NN1 = 6-((quinolin-2-ylmethylene)amino)-2H-chromen-2-one

Section 2.1 describes the synthesis of the complex in sufficient detail. Moreover, as far as can be understood from the work, this complex had previously been obtained by the same authors and published in Molecules 2020, 25(14), 3183. In this case, it is enough to indicate that the complex was synthesized using a known method and provide a reference. It is only necessary to leave a description of the modification that made the synthesis more efficient.

The authors in section 2.1 provide a link to Figure 1; it is not shown in the manuscript. If this figure repeats the figure from Molecules 2020, 25(14), 3183, I propose to place it in the supplementary; if it contains significant modifications, it should be included in the main part of the article.

Many references do not list pages or article numbers, for example: 13, 20, 21, 25, 27, 29, 30 etc.

After taking into account the comments and revising the synthetic part, the article can be published.

Author Response

  • The abstract contains only the abbreviated formula of the copper complex -[Cu(NN1)2](ClO4), it is necessary to give the decoding of the ligand when first mentioned - NN1 = 6-((quinolin-2-ylmethylene)amino)-2H-chromen-2-one

Response: the decoding of the ligand was added:

Abstract: Piscirickettsia salmonis is the pathogen that most affects the salmon industry in Chile. Large quantities of antibiotics have been used to control it. In search of alternatives, we have developed [Cu(NN1)2]ClO4, where NN1 = 6-((quinolin-2-ylmethylene)amino)-2H-chromen-2-one. The antibacterial capacity of [Cu(NN1)2]ClO4 was determined. Subsequently, the effect of the administration of [Cu(NN1)2]ClO4 on the growth of S. salar, modulation of the immune system, and the intestinal microbiota was studied. Finally, the ability to protect against a challenge with P. salmonis was evaluated. The results obtained showed that the compound has a MIC between 15 and 33.9 mg/mL in four isolates. On the other hand, the compound did not affect the growth of the fish, however, an increase in the transcript levels of IFN-γ, IL-12, IL-1β, CD4, lysozyme, and perforin was observed in fish treated with 40 mg/g of fish. Furthermore, modulation of the intestinal microbiota was observed, increasing the genera of beneficial bacteria such as Lactobacillus and Bacillus, and also potential pathogens such as Vibrio, and Piscirickettsia. Finally, the treatment increased survival in fish challenged with P. salmonis by more than 60%. These results demonstrate that the compound is capable of protecting fish against P. salmonis, probably by modulating the immune system and the composition of the intestinal microbiota.

  • Section 2.1 describes the synthesis of the complex in sufficient detail. Moreover, as far as can be understood from the work, this complex had previously been obtained by the same authors and published in Molecules 2020, 25(14), 3183. In this case, it is enough to indicate that the complex was synthesized using a known method and provide a reference. It is only necessary to leave a description of the modification that made the synthesis more efficient.

Response: Indeed, the complex was previously synthesized and published in Molecules 2020, 25(14), 3183, as indicated in the manuscript, so the referee's request is accepted and only the description of the modifications is left.

2.1. Synthesis of Cu (I) coordination complex

Previously, in our laboratory, we have synthesized the coordination complex [Cu(NN1)2]ClO4, where NN1 = 6-((quinolin-2-ylmethylene)amino)-2H-chromen-2-one, a ligand derivate from natural product coumarin 1-benzopyran-2-one (Aldabaldetrecu et al., 2020). In this paper, we account for the changes made in the experimental procedure, to improve the process. It is so, on this occasion, the synthesis was easily by template condensation method from equimolar amounts of reagents as shown in Figure 1, which allowed the reaction time to be reduced to 1 hour, increasing the reaction yield to 90%, at room temperature and without requiring pressure systems, substantially improving the energy and economic cost of the process.

  • The authors in section 2.1 provide a link to Figure 1; it is not shown in the manuscript.

Response: Figure 1 has been added to the paper

  • Many references do not list pages or article numbers, for example: 13, 20, 21, 25, 27, 29, 30 etc.

Response: References have been reviewed and completed

Round 2

Reviewer 1 Report

Comments and Suggestions for Authors

Thanks to the authors for the detailed responds to the comments. Unfortunately, none of the responses are included in revised version of the article. Please incorporate detailed information from Response 2 into the part Introduction. Response 3 should be part of Conclusion.

Author Response

Referee 1

Thanks to the authors for the detailed responds to the comments. Unfortunately, none of the responses are included in revised version of the article. Please incorporate detailed information from Response 2 into the part Introduction. Response 3 should be part of Conclusion.

Response:

We thank the referee for his comments and respond to his last suggestion.

The phrase “and that when administering 60 μg/g of fish to rainbow trout it generates protection against F. psychrophilumwas modified in the introduction.

The phrase “The results obtained in this work showed the different effects of the compound, such as the antibacterial activity on P. salmonis, and the ability to modulate the immune response and the intestinal microbiota in Atlantic salmon. These multiple effects of the compound could be the mechanisms by which it generates protection against P. salmonis in Atlantic salmon. These are promising results for this compound as an alternative treatment to the Piscirickettsiosis” was added in the introduction.

The phrase “ However, it is necessary to continue carrying out tests and studies to increase the characterization of the effect of the compound, such as the evaluation of toxicity on other aquatic organisms, its accumulation on the seabed, synergism effect with antibiotics, among other experiments” was added in the conclusion.
